# OpenReview forum: "HarmLevelBench: Evaluating Harm-Level Compliance and the Impact of Quantization on Model Alignment"
_NeurIPS.cc/2024/Workshop/SafeGenAi — SafeGenAi Poster_

### Official Review · Reviewer_XR6Z · 2024-10-08
**The evaluations are only limited to Vicuna 13B model**

**Rating:** 5
**Confidence:** 4

**Review:**

In this work, the authors first propose a new dataset, named as HarmLevelBench, which contains 7 harmful topics and 8 severity levels. Then on this dataset, they conduct experiments to evaluate the impact of model quantization on the adversarial defense/attack. Despite the fact that very few work considered model quantisation in adversarial attack, the evaluations are limited to Vicuna 13B v1.5 the single model. And this model is relatively easier to be jailbroken than llama2/llama3. Moreover, an explanation / in-depth analysis of why model quantisation has such impacts on adversarial attack is absent.

---

### Official Review · Reviewer_6rjm · 2024-10-09

**Rating:** 6
**Confidence:** 3

**Review:**

This paper addresses the problem of benchmarking LLMs for vulnerabilities. The authors' contribution includes the creation of a new dataset designed to assess the harmfulness of model outputs across multiple harm levels, as well as a focus on fine-grained harm-level analysis. The authors claim that state-of-the-art datasets for evaluating LLMs often lack sufficient formulation structure of queries and fail to provide a nuanced assessment of the varying degrees of harmfulness in the queries. In this context, the authors propose the HarmLevelBench dataset, which covers a diverse range of 7 potentially harmful topics.

The paper proposes a comprehensive set of experiments and comparisons with other benchmarking datasets. They also use their dataset to investigate how quantization techniques, such as AWQ and GPTQ, affect the tuning and robustness of models, revealing trade-offs between robustness and vulnerability.

- The quality of the paper is good, the methodology is interesting and well explained
- The paper is well-written and well-structured
- The paper can have an impact if their benchmark is used by the community.

---

### Official Review · Reviewer_DtDj · 2024-10-09

**Rating:** 3
**Confidence:** 5

**Review:**

Summary:

This paper introduces a new dataset HarmLevelBench with 7 harmful topics and 8 distinct levels of severity. The authors also explore how quantization techniques influence the alignment and robustness of models on Vicuna13B.

Pros
- Harm level is important for fine-grained analysis.
- Investigation of quantization impact is interesting.

Cons
- The focus of this paper seems to be disjointed. Is it a dataset paper, or an empirical investigation paper?
- If a dataset paper, then the author should clearly describe how to obtain the data and how to annotate data (e.g. the harm level). They also should provide statistics of the introduced HarmLevelBench.
- For the dataset, why does the formulation structure of queries (question template) matter, which is pointed out by authors as one main limitation?  How do different formats influence the evaluation? If the author cannot provide evidence for this point, this motivation would make no sense.
- Furthermore, existing datasets also provide semantic categories such as HarmBench [1].
- If an empirical investigation paper, the authors should provide a more comprehensive evaluation. The evaluations on only Vicuna13B v1.5 and its AWQ and GPTQ variants are weak.
- Besides, the relationship between harmfulness level and jailbreaking complexity is not straightforward. The annotation of the harm level could be noisy, and it is even hard for humans to give a quantitative value for query harmfulness without specific rubric and detailed guidance. Also, how is the jailbreaking complexity obtained? Why GCG is more complex than AutoDAN-GA? It is not convincing without details and the conclusion could even make no sense.

Reference:

[1] Mazeika, M., Phan, L., Yin, X., Zou, A., Wang, Z., Mu, N., Sakhaee, E., Li, N., Basart, S., Li, B. and Forsyth, D., 2024. Harmbench: A standardized evaluation framework for automated red teaming and robust refusal. ICML.